# Emotions and Food Consumption: Emotional Eating Behavior in a European Population

**DOI:** 10.3390/foods12040872

**Published:** 2023-02-17

**Authors:** Marija Ljubičić, Marijana Matek Sarić, Ivo Klarin, Ivana Rumbak, Irena Colić Barić, Jasmina Ranilović, Boris Dželalija, Ana Sarić, Dario Nakić, Ilija Djekic, Małgorzata Korzeniowska, Elena Bartkiene, Maria Papageorgiou, Monica Tarcea, Maša Černelič-Bizjak, Dace Klava, Viktória Szűcs, Elena Vittadini, Dieuwerke Bolhuis, Raquel P. F. Guiné

**Affiliations:** 1Department of Health Studies, University of Zadar, 23000 Zadar, Croatia; 2General Hospital Zadar, 23000 Zadar, Croatia; 3Faculty of Food Technology and Biotechnology of Zagreb, 10000 Zagreb, Croatia; 4Podravka Ltd., Research & Development, 48000 Koprivnica, Croatia; 5School of Medicine, Catholic University of Croatia, 10000 Zagreb, Croatia; 6Faculty of Agriculture, University of Belgrade, 11000 Belgrade, Serbia; 7Department of Functional Food Products Development, Wroclaw University of Environmental and Life Sciences, 51-630 Wrocław, Poland; 8Department of Food Safety and Quality, Lithuanian University of Health Sciences, 44307 Kaunas, Lithuania; 9Department of Food Science and Technology, International Hellenic University, 57400 Thessaloniki, Greece; 10Department of Community Nutrition and Food Safety, George Emil Palade University of Targu Mures, 540141 Targu Mures, Romania; 11Faculty of Health Sciences, University of Primorska, 6310 Izola, Slovenia; 12Department of Food Technology, Faculty of Food Technology, Latvia University of Agriculture, LV 3001 Jelgava, Latvia; 13Hungarian Chamber of Agriculture, Directorate of Food Industry, H-119 Budapest, Hungary; 14School of Biosciences and Veterinary Medicine, University of Camerino, 62032 Camerino, Italy; 15Food Quality and Design Group, Wageningen University & Research, Bornse Weilanden 9, 6708 WG Wageninge, The Netherlands; 16Centro de Estudos de Recursos Naturais, Ambiente e Sociedade, Research Centre, Polytechnic Institute of Viseu, 3504-510 Viseu, Portugal

**Keywords:** emotional eating behavior, emotional motivation, motivation for health behavior, stress-related eating, negative emotions, lifestyle

## Abstract

Emotion can reflect in the perception of food consumption. An increase in food intake during emotional and psychological conditions may have a negative impact on human health. The aim of this cross-sectional study was to determine the associations between food consumption, emotional eating behavior, and emotional conditions such as stress, depression, loneliness, boredom eating, maintaining vigilance and alertness, and emotional food consolation. We used a Motivations for Food Choices Questionnaire (Eating Motivations, EATMOT) to determine the emotional aspects of food consumption in 9052 respondents living in 12 European countries between October 2017 and March 2018. Ordinal linear regression was used to identify the associations between the emotional eating behavior and emotional conditions such as stress, depression, loneliness, emotional consolation, and reasons to improve physical and psychological conditions. The regression models confirmed the associations between food consumption, emotional conditions, and emotional eating behavior. Associations were found between the emotional eating behavior and stress (odds ratio (OR) = 1.30, 95% confidence interval (CI) = 1.07–1.60, *p* = 0.010), depressive mood (OR = 1.41, 95% CI = 1.40–1.43, *p* < 0.001), loneliness (OR = 1.60, 95% CI = 1.58–1.62, *p* < 0.001), boredom (OR = 1.37, 95% CI = 1.36–1.39, *p* < 0.001), and emotional consolation (OR = 1.55, 95% CI = 1.54–1.57, *p* < 0.001). Emotional eating was associated with an effort to improve physical and psychological conditions, such as controlling body weight (OR = 1.11, 95% CI = 1.10–1.12, *p* < 0.001), keeping awake and alert (OR = 1.19, 95% CI = 1.19–1.20, *p* < 0.001) and consumption to feel good (OR = 1.22, 95% CI = 1.21–1.22, *p* < 0.001). In conclusion, emotions might provoke emotional eating behavior. The appropriate way to handle stress, depression, or other emotional states is important in conditions of being emotionally overwhelmed. The public should be educated on how to handle different emotional states. The focus should be moved somehow from emotional eating and the consumption of unhealthy food to healthy lifestyle practices, including regular exercise and healthy eating habits. Thus, it is necessary to halt these negative health effects on human health through public health programs.

## 1. Introduction

Emotion as a reaction to the cognitive assessment of stimuli from the environment is an integral part of human existence, behavior, and functioning [1,2,3]. Emotions are closely related to food choice and the ritual of feeding as a social activity but also to different health and emotional outcomes. Emotions are related to mood, yet moods are stronger and last longer [4]. Studies have confirmed the connection between mood or emotional conditions and food choice and different eating disorders, as well as the perception that food can be a way to fill a void that may arise under conditions such as loneliness, sadness, depression, social isolation, and other emotional moods caused by stress, excitement and tension, conflicted social relationships, and other stressful life events [5]. These events can occur at any time throughout one’s life, but studies have shown that incidences of heightened emotionality are more pronounced in early adolescence; however, when during childhood, they have a negative impact on eating patterns [6,7,8]. Although less common than in younger people, emotional eating occurs in older people [6]. Women are more likely than men to develop eating disorders [9]. Thereby, social media may have a strong influence on the relationship between mood and eating and the development of eating patterns [10]. The ways of dealing with different emotions may be projected through food habits expressed as emotional eating. Moreover, an increase in food intake during emotional and psychological conditions such as stress may have a negative impact on health. An unbalanced diet may foster many chronic diseases, such as metabolic syndrome, obesity, diabetes mellitus, hypercholesterolemia, high blood pressure, ischemic heart disease, and stroke [11,12,13,14,15].

Certain segments of life are almost impossible to imagine without a certain level of excitement and tension [16,17]. This tension includes usual daily stress and accompanying emotions related to stressful perceptions. The presence of stress as an integral part of daily life is almost inevitable [18]. The way a person copes with stressful circumstances affects his functioning [18]. Although stress can be viewed in a positive light, having a stimulating effect and encouraging certain activities, it is mainly considered a negative stimulus. Usual daily stress can be intensified and/or cause emotional responses in the form of psychological reactions. Studies have confirmed the association between stress perception, emotional reaction, depression, anxiety, social isolation, and loneliness [12,19,20]. These conditions can be reflected in the perception of food consumption, resulting in decreased or increased food intake [15,21,22].

Stress-related eating is “defined as trying to make oneself feel better by eating or drinking in a stressful situation” [23]. Often, these intakes are associated with unhealthy comfort eating and eating without hunger. In this state, during eating, we usually do other activities, and the feeding process is out of our conscious control. Additionally, certain behaviors encouraged by hedonic motivation may manifest in a felt desire to engage in behaviors that bring pleasure, such as eating sweet foods [24]. All of these behaviors can be explained as responses to emotions rather than hunger that lead to excessive calorie consumption [25]. This state in which a person is unable to distinguish hunger from emotional arousal refers to emotional eating [26].

Emotional eating is described as a “tendency to eat in response to negative emotions with the chosen foods being primarily energy-dense and palatable ones” [15]. This condition can be caused by different factors, such as coping with stress and other negative emotions such as depression and feeling loneliness [15,27]. Many people respond to stress by increasing their food intake because food is often a comfort in these circumstances [28]. Moreover, emotional eating is the overlapping mechanism between reward circuitry, cognitive control, and emotions. Some pathophysiology mechanisms, such as an energy imbalance in the hypothalamus and its relationship with ghrelin (“hunger hormone”) and leptin (“body’s satiety signal”), have a significant impact on mood disorders [29]. Studies have confirmed that the consumption of highly palatable foods relates to the pathways of the limbic system to mediate motivated behaviors, which explains why emotional eating overcomes cognitive control mechanisms [30,31].

From a psychological point of view, the consumption of food as a form of comfort can alleviate the stress reaction and thus act as a preventive measure against stress reactions. However, difficulty arises when a person under stress consumes food with a poor nutritional composition [28]. When stressed, people often eat unhealthy foods high in salt, sugar, and fat, typically in high quantities, and often in the absence of hunger [21,28,32]. Additionally, studies have confirmed negative associations between emotional eating and basic human needs, such as physical needs, safety, love, belongingness, self-actualization, and self-esteem [33].

Studies have explained the associations between negative emotions and unhealthy food consumption [27]. Likewise, unhealthy eating patterns can cause mood disorders, creating a vicious circle of unhealthy eating, behavior, and bad feelings and increasing the consumption of unhealthy food. This behavior can be a result of interference by emotions and the mutual relationship of regulatory processes. Namely, “emotions may regulate eating, and eating may regulate emotions” [22]. Despite evidence finding emotional motivations for eating in stress perceptions, depression, loneliness, and using food as emotional consolation, the results from many studies are mixed [5,15,21,22,28,34,35]. Eating while under stress may have anxiolytic properties, but there is negligible empirical evidence for this [21]. Studies have also shown that positive emotions may influence food consumption the same as negative emotions [5,27]. However, many studies have indicated that individuals tend to eat more high-energy foods when confronted with negative emotions [5]. This can cause an accumulation of energy, causing the consumption of more unhealthy drinks and foods, thereby increasing body weight and resulting in many chronic diseases [4]. According to the United States Centers for Disease Control and Prevention (CDC), many chronic diseases are caused by unhealthy food intake and poor nutrition, which has become a serious global health concern [27,36].

The aim of this study was to determine the associations between food consumption, eating behavior, and emotional conditions such as stress, depression, loneliness, boredom eating, maintaining vigilance and alertness, and emotional food consolation. We hypothesize that this emotional eating behavior (EEB) is associated with some unhealthy lifestyle habits, demographic characteristics, motivation for healthy behavior (MHB), and emotional conditions. Triggered by negative emotions, EEB may reflect the increased consumption of unhealthy foods and negative lifestyle habits with negative consequences on health outcomes.

## 2. Materials and Methods

### 2.1. Participants

A cross-sectional survey was conducted on a nonprobabilistic convenience sample of 9052 respondents living in 12 European countries: Croatia (1538; 17.0%), Greece (498; 5.5%), Hungary (500; 5.5%), Italy (541; 6.0%), Latvia (636; 7.0%), Lithuania (507; 5.6%), the Netherlands (521; 5.8%), Poland (586; 6.5%), Portugal (1314; 14.5%), Romania (821; 9.1%), Serbia (498; 5.5%), and Slovenia (1093; 12.1%). The respondents were recruited from shopping centers, universities, downtown areas, and rural areas through advertisements or by word of mouth. To be eligible for inclusion, the respondents needed to be 18 years or older and had to voluntarily agree to fill out the questionnaire. The exclusion criteria were tourists or students from other countries and participants who had not completed the questionnaire. The study was voluntary and there was no compensation for participation.

This study was approved by the Ethical Committee of the Polytechnic Institute of Viseu (registration number 04/2017) with additional approval of the ethics committees of each participating country before the start of data collection. The study took place between October 2017 and March 2018 and was conducted in accordance with the ethical standards of the Declaration of Helsinki.

### 2.2. Questionnaire

We used a paper-and-pencil Motivations for Food Choices Questionnaire (Eating Motivations, EATMOT) to determine the emotional aspects of food consumption in 12 different European countries. The final version of the working instrument had already been worked, tested, and validated for a sample of the Portuguese population [37]. All of the other countries received the working instrument in English and their own language without any changes in the questions, any more or fewer questions, or any more options, ensuring that all of the results/answers were compatible regardless of country. To translate the questionnaire from English into the different languages, the back-translation process was used for better accuracy; translation from English into their native language was conducted by some experts and then from their native language into English again by different people. The final version of the questionnaire was sent to Portugal to the project manager for verification and approval (not the exact words because the project manager was not able to read most of the participants’ native languages but to see if anything in the original structure was changed). The validation of the questionnaire was described in our previous papers [38,39]. The validated instrument guarantees confidence and confirms that the information obtained through this instrument is compatible with the population of the selected participating countries [39].

For each of the participating countries, we conducted a small pilot test on 10 respondents from different age groups before administering the questionnaire to ensure that it was understandable and would yield results that fell within the expected range. Filling out the questionnaire took an average of 15 min. Both the location and the time needed to complete the questionnaire were up to the participants. The researchers received the completed questionnaires by courier or in person.

The questionnaire contained three components: sociodemographic and lifestyle habits data (18 items), motivation for health behavior (MHB) data (10 items), and emotional eating behavior (EEB) data (9 items). 

The sociodemographic data included age, gender, education level, country and place, marital status, and employment status. The lifestyle habits included anthropometric data (weight in kilograms, height in meters, and body mass index (BMI = kg/m^2^)), physical exercise, and time in hours per day spent watching television or in front of a computer. The participants’ BMI was calculated using the following formula: BMI = body weight (kg)/(body height (m^2^) (BMI = kg/m^2^)) from self-assessed anthropometric data (weight in kilograms and height in meters).

The items about MHB and EEB were defined by a Likert scale from 1 to 5 (1 = totally disagree, 2 = disagree, 3 = neither agree nor disagree, 4 = agree, and 5 = strongly agree). MHB involved questions about the hygiene and safety of food, a healthy and balanced diet, a diet rich in vitamins and minerals, a low-fat diet, a low level of cholesterol and sugar, an additive-free diet, and a diet without processed foods or genetically modified organisms. Overall MHB was the sum of all 10 responses (ranging from 10 to 50); a higher score denoted a greater level of motivation for healthy nutrition behavior.

The EEB data included questions about the emotional conditions of food consumption: coping with stress, consuming sweets when depressed, perception of food as emotional consolation, food consumption when bored (boredom eating), food as consolation when feeling lonely, making oneself feel good, keeping oneself awake and alert, helping to control weight, and helping to relax. Overall EEB was the sum of all nine responses (ranging from 9 to 45); a higher score denoted a greater level of food consumption in various emotional conditions.

### 2.3. Statistical Methods

Statistical analysis was conducted using SPSS 26.0 (IBM, Armonk, NY, USA). To assess the data distribution, we used the Kolmogorov–Smirnov normality test. Due to non-normal distribution, the median and interquartile range for numerical variables were used, while percentages and absolute numbers were used to describe categorical variables. To analyze the difference between groups, we used the chi-square and Kruskal–Wallis tests with the Mann–Whitney U test as post hoc.

We created multivariate regression models to assess the association between the emotional conditions of food consumption, emotional eating behavior, motivation for healthy nutrition behavior, and predictors.

The regression analysis included ordinal regression models. Several ordinal regression models were used to access the associations between the ordinal dependent variables: consuming food as a way to cope with stress, consuming sweets when depressed, perception of food as emotional consolation, food consumption when bored, food consolation when lonely, making oneself feel good, keeping oneself awake and alert, helping to control weight, and helping to relax. The independent variables were countries as studies groups (respondents of Croatia were the reference group), sociodemographic variables (age (elderly ≥ 66 years were the reference group), gender (males were the reference group), marital status (married was the reference group), education (university was the reference group), residential environment (urban was the reference group), and employment (employment was the reference group)), profession (nutrition, food, agriculture, sport, psychology, or health; another profession not related to the list was the reference group), following a healthy diet (yes was the reference group), category of BMI (obesity ≥ 30 was the reference group), physical activity (weekly was the reference group), sitting in front of a television or computer, overall health motivation, and emotional eating behavior. Statistically significant values were those with *p* < 0.05.

## 3. Results

### 3.1. Demographic Characteristics of the Study Groups

All of the sociodemographic characteristics of the participants, according to the country of residence, were different by age, gender, marital status, education, residential environment, and employment (Table 1). The average age of all of the participants was 33 years (the interquartile range was 23). The Hungarian participants were the oldest (Mdn = 43.0; IQR = 18.0), while the participants from Serbia were the youngest (Mdn = 23.0; IQR = 7.0). The female gender was dominant in all countries, with the male gender being the highest in the Hungarian sample at 49.4% (*p* < 0.001). The Greek participants reported higher education (86.5%), while the Hungarian participants reported the lowest (32.2%) (*p* < 0.001). Urban populations were dominant, ranging from 68.4% for Italy to 95.4% for the Netherlands. The Italian participants reported a higher marriage rate, while the Serbian participants reported higher single, divorced, and widowed rates. The Latvian participants reported a higher level of employment, while the Serbian participants reported higher unemployment, retirement, and student levels. Most of the participants were from professions that do not include food, nutrition, agriculture, sport, psychology, or health. However, healthcare workers were the most represented in Romania (39.2%), Latvia (30.2%), and Croatia (22.8%), while food workers were highly represented in Lithuania (31.8%), Latvia (28.6%), and Poland (24.1%). Other professions were represented less than health professionals (Table 1).

### 3.2. Lifestyle Habits and Health Motivation Eating Behavior

Following a healthy diet ranged from Mdn = 3.0–4.0 (IQR = 0.0–1.0). In almost all countries, the participants had a normal BMI except the Hungarian participants with a BMI Mdn = 25.9 (IQR = 7.1), mostly distributed in the overweight (33.2%) and obese (24.0%) categories. The Hungarian participants reported the lowest frequency of exercise (52.6%), while the highest was in the Slovenian participants (81.9%). The most hours per day in front of the television or a computer was reported by the Portuguese and Latvian participants (Mdn = 6.0; IQR = 5.0), and the least was reported by the Slovenian participants (Mdn = 2.0; IQR = 3.0). Motivation for positive health behavior was the highest in the Portuguese participants (Mdn = 38.0; IQR = 7.0), and the lowest was in the Dutch (Mdn = 30.0; IQR = 7.0) and Romanian participants (Mdn = 30.0; IQR = 8.0) (Table 1).

### 3.3. Emotional Conditions of Food Consumption

Bivariate analysis showed that in comparison to the other European countries, the highest frequency of comfort eating to cope with stress was reported by the Lithuanian respondents (Mdn = 3.0; IQR = 2.0, Mean Rank = 5399.1, *p* < 0.001) and the lowest by the Hungarian respondents (Mdn = 2.0; IQR = 2.0, Mean Rank = 3324.5, *p* < 0.001). Similar results were obtained for boredom eating, helping to relax, and food consolation when lonely. Additionally, the Lithuanian population reported a higher frequency of eating sweets when depressed and the perception of food as an emotional consolation (*p* < 0.001 for all) (Table 2). The Latvian population reported eating food to keep alert and awake and to make themselves feel well more than the other countries (Mdn = 4.0; IQR = 2.0, Mean Rank = 5575.1, *p* < 0.001). The Slovenian respondents reported a higher frequency of eating food to control weight (Mdn = 4.0; IQR = 1.0, Mean Rank = 5017.2, *p* < 0.001), while the Serbian respondents reported the lowest frequency (Mdn = 2.0; IQR = 2.0, Mean Rank = 3278.6, *p* < 0.001) (Table 2).

Overall EEB was the highest among the Lithuanian respondents (Mdn = 30.0; IQR = 8.0, mean rank = 5800.0, *p* < 0.001) and the lowest among the Hungarian respondents (Mdn = 23.0; IQR = 10.0, mean rank = 3453.4, *p* < 0.001) (Table 2 and Figure 1 and Appendix A). Statistically significant differences were found between European countries in emotional eating behavior during different emotional states except between Greece and Italy (Appendix A).

### 3.4. Consuming Food as a Way of Coping with Stress and Depression 

The ordinal regression models confirmed the associations between emotional statuses such as stress, depressive mood, loneliness, boredom, and emotional consolation as reasons for food consumption and the predictors (Table 3). In comparison to the Croatian respondents, higher odds for consuming food as a way of coping with stress were recorded in the Slovenian respondents (odds ratio (OR) = 1.90; 95% confidence interval (CI) = 1.62–2.23; *p* < 0.001), the Serbian respondents (OR = 1.62; 95% CI = 1.32–1.98; *p* < 0.001), the Dutch respondents (OR = 1.40; 95% CI = 1.15–1.72; *p* = 0.001), the Italian respondents (OR = 1.40; 95% CI = 1.15–1.69; *p* = 0.001), and the Greek respondents (OR = 1.30; 95% CI = 1.07–1.60; *p* = 0.010). Lower odds were recorded in the Hungarian respondents (OR = 0.56; 95% CI = 0.45–0.69; *p* < 0.001) and the Latvian respondents (OR = 0.83; 95% CI = 0.69–1.00; *p* = 0.049) compared to the Croatian respondents. Agriculture (OR = 1.30; 95% CI = 1.06–1.61; *p* = 0.013) and health professionals (OR = 1.15; 95% CI = 1.02–1.29; *p* = 0.020) had higher odds of consuming food as a way of coping with stress when compared to the other professions. Higher odds of stress being the reason for food consumption were recorded in those subjects who never or rarely followed a healthy diet (OR = 1.22; 95% CI = 1.08–1.38; *p* = 0.001, compared to yes) and were overweight (OR = 1.42; 95% CI = 1.41–1.44; *p* < 0.001, compared to obesity), while those subjects who were underweight (OR = 0.65; 95% CI = 0.50–0.83; *p* = 0.001) or of normal weight (OR = 0.97; 95% CI = 0.96–0.98; *p* < 0.001) had lower odds when compared to the obese respondents. An increase in emotional reasons for food consumption was associated with an increase in the odds of coping with stress being the reason for food consumption (OR = 1.30; 95% CI = 1.07–1.60; *p* = 0.010). Associations between consuming food as a way of coping and sociodemographic predictors were not recorded.

Eating sweets when depressed was recorded in the Hungarian (OR = 1.64; 95% CI = 1.34–2.02; *p* < 0.001), Slovenian (OR = 1.45; 95% CI = 1.24–1.70; *p* < 0.001), Greek (OR = 1.28; 95% CI = 1.05–1.57; *p* = 0.015), and Italian respondents (OR = 1.27; 95% CI = 1.05–1.54; *p* = 0.015) compared to the Croatian respondents. Higher odds of eating sweets when depressed were recorded in the rural respondents (OR = 1.19; 95% CI = 1.06–1.34; *p* = 0.003, compared to the urban respondents) and subjects who never or rarely followed a healthy diet (OR = 1.12; 95% CI = 0.99–1.27; *p* = 0.064, compared to yes). An increase in emotional reasons for food consumption was associated with an increase in the odds of eating sweets to cope when depressed (OR = 1.41; 95% CI = 1.40–1.43; *p* < 0.001) (Table 3). 

### 3.5. Finding Emotional Consolation in Food Consumption

In comparison to Croatia, almost all countries recorded higher odds for consoling through food consumption when lonely, especially those respondents from the Netherlands (OR = 2.74; 95% CI = 2.22–3.38; *p* < 0.001) and Lithuania (OR = 2.33; 95% CI = 1.88–2.88; *p* < 0.001) (Table 3). Higher odds of seeking consolation through eating food when lonely were recorded in women (OR = 1.14; 95% CI = 1.03–1.26; *p* = 0.014), subjects without a university education (OR = 1.13; 95% CI = 1.02–1.24; *p* < 0.001), single, divorced, or widowed respondents (OR = 1.14; 95% CI = 1.02–1.26; *p* = 0.015), subjects who never or rarely follow a healthy diet (OR = 1.12; 95% CI = 0.99–1.27; *p* = 0.064), and those who never exercise (OR = 1.15; 95% CI = 1.04–1.27; *p* = 0.004). An increase in motivation for health behaviors was associated with a decrease (OR = 0.94; 95% CI = 0.93–0.95; *p* < 0.001), while emotional reasons for food consumption were associated with an increase in the odds of food consumption as consolation when lonely (OR = 1.60; 95% CI = 1.58–1.62; *p* < 0.001).

A similar finding was also recorded for boredom eating. The respondents from the Netherlands, younger subjects, females, rural subjects, and subjects who never follow a healthy diet and never exercise recorded higher odds. An increase in motivation for healthy nutrition behaviors was associated with a decrease in the odds (OR = 0.94; 95% CI = 0.93–0.95; *p* < 0.001), while an emotional reason for food consumption was associated with an increase in the odds of eating food when bored (OR = 1.37; 95% CI = 1.36–1.39; *p* < 0.001) (Table 3).

Lower odds of finding emotional consolation in food were recorded in almost all countries when compared to Croatia. Higher odds were recorded in female respondents (OR = 1.22; 95% CI = 1.11–1.35; *p* < 0.001), unemployed respondents (OR = 1.14; 95% CI = 1.02–1.27; *p* = 0.025), health professionals (OR = 1.35; 95% CI = 1.20–1.53; *p* < 0.001), and subjects who never exercise (OR = 1.15; 95% CI = 1.05–1.26; *p* = 0.004). Moreover, the odds for higher emotional consolation were associated with higher emotional motivation (OR = 1.55; 95% CI = 1.54–1.57; *p* < 0.001), while lower odds for emotional consolation via eating were associated with higher motivation for healthy nutrition behavior (OR = 0.95; 95% CI = 0.94–0.95; *p* < 0.001) (Table 3).

### 3.6. Improving Physical and Psychological Conditions through Food Consumption

Higher odds of controlling body weight by food were recorded in the Portuguese (OR = 1.53; 95% CI = 1.32–1.78; *p* < 0.001) and Slovenian respondents (OR = 1.29; 95% CI = 1.10–1.50; *p* = 0.001) compared to the Croatian respondents. In comparison to obese respondents with a BMI of ≥30, those respondents who were of normal weight (OR = 1.34; 95% CI = 1.15–1.55; *p* < 0.001) and overweight (OR = 1.26; 95% CI = 1.08–1.46; *p* = 0.003) had higher odds of consuming food in a way that helps them control their body weight (Table 4).

Higher odds of often consuming foods that keep oneself awake and alert (such as coffee, Coca-Cola, and energy drinks) were recorded in those respondents from Romania (OR = 1.96; 95% CI = 1.67–2.30; *p* < 0.001), Hungary (OR = 1.84; 95% CI = 1.52–2.23; *p* < 0.001), Slovenia (OR = 1.73; 95% CI = 1.43–2.09; *p* < 0.001), and Latvia (OR = 1.21; 95% CI = 1.01–1.44; *p* = 0.037) compared to the Croatian respondents. Those of a younger age had higher odds of consuming such food (OR = 1.69; 95% CI = 1.30–2.21; *p* < 0.001). Health professionals had higher odds of consuming foods to keep themselves awake and alert (OR = 1.28; 95% CI = 1.15–1.43; *p* < 0.001). A lower body mass index was associated with higher odds of consuming foods to keep oneself awake and alert (BMI < 18.5 (OR = 1.59, 95% CI = 1.26–2.01; *p* < 0.001); BMI = 18.5–24.9 (OR = 1.30; 95% CI = 1.13–1.50; *p* < 0.001); BMI = 25.0–29.9 (OR = 1.17; 95% CI = 1.01–1.36; *p* = 0.034). Similar findings were found for those who never exercise (OR = 1.30; 95% CI = 1.19–1.41; *p* < 0.001) and subjects who spend more hours in front of a computer or television (OR = 1.02; 95% CI = 1.00–1.03; *p* = 0.014).

In comparison to Croatia, eating for relaxation was recorded in Portuguese (OR = 2.86; 95% CI = 2.47–3.30; *p* < 0.001), Serbian (OR = 1.58; 95% CI = 1.31–1.92; *p* < 0.001), and Polish respondents (OR = 1.47; 95% CI = 1.22–1.76; *p* < 0.001). A lower body mass index and sitting in front of a computer or television were associated with higher odds of food consumption for relaxation comfort. The other predictors were associated with lower odds of consuming food for relaxation (Table 3). Similar results were found when food was being consumed to make oneself feel good. 

An increase in motivation for health behaviors was associated with higher odds of increasing the consumption of food in a way that helps to control body weight (OR = 1.19; 95% CI = 1.18–1.21; *p* < 0.001) and consuming food to feel good (OR = 1.04; 95% CI = 1.04–1.05; *p* < 0.001) but with lower odds of consuming foods to keep oneself awake and alert (OR = 0.95; 95% CI = 1.94–0.96; *p* < 0.001). Emotional reasons for food consumption were associated with controlling body weight (OR = 1.11; 95% CI = 1.10–1.12; *p* < 0.001), keeping oneself awake and alert (OR = 1.19; 95% CI = 1.19–1.20; *p* < 0.001), and the perception that food makes oneself feel good (OR = 1.22; 95% CI = 1.21–1.22; *p* < 0.001) (Table 4). Higher odds for controlling body weight by food were recorded in the Portuguese (OR = 1.53; 95% CI = 1.32–1.78; *p* < 0.001) and Slovenian respondents (OR = 1.29; 95% CI = 1.10–1.50; *p* = 0.001) compared to the Croatian respondents. In comparison to obese respondents with a BMI of ≥30, those respondents who were of normal weight (OR = 1.34; 95% CI = 1.15–1.55; *p* < 0.001) and overweight (OR = 1.26; 95% CI = 1.08–1.46; *p* = 0.003) had higher odds of consuming food in a way that helps them control their body weight (Table 4).

## 4. Discussion

This study aimed to identify associations between food consumption and emotional eating behavior arising from emotional conditions such as stress, depression, loneliness, the need to maintain alertness, relaxation, and making oneself feel good. Our results indicate that individuals with higher emotional motivation and eating behavior conditioning are at risk of eating under the conditions of stress, depression, loneliness, and boredom. EEB was associated with eating to keep oneself awake and alert, to make oneself feel good, and to help control weight. This can be explained by the fact that food consumption and emotional eating may be influenced by different environmental and individual determinants [25]. Additionally, in our study, in contrast to the motivations behind health behaviors, higher emotional motivation under conditions when stressed, depressed, bored, and lonely may result in eating as a form of emotional consolation and in negative health outcomes. Various studies have confirmed these results. For example, stress contributes to an unhealthy lifestyle, fast food consumption, and emotional eating [16,28,35,40]. Depression is associated with stress and a bad mood, as well as with loneliness and social isolation, which are also associated with emotional eating [12]. Additionally, studies have noted an association between food intake and immediate mood improvement, and have confirmed the mediation effect of tastiness between food intake and mood improvement [34].

Focusing on our results of the associations between emotional conditions, a priori stress perceptions, and food consumption, we found different results. When compared to the Croatian respondents, the Slovenian, Serbian, Dutch, Italian, and Greek respondents were more likely to eat food as a way of coping with stress, while the Hungarian and Latvian respondents were less likely. These differences in the odds among countries in terms of the contribution of stress to food consumption can be explained by different residential environments, different levels of life satisfaction, lifestyle, and culture but also by different health and social statuses and standards of living [41,42,43]. For example, compared to other countries, the Slovenian sample was dominated by women and respondents who eat food to help control their weight. In Serbia, the majority of the population was unemployed, and these respondents eat foods that help them control their weight less so than others. It is possible that certain sociodemographic factors, such as unemployment, contribute to more significant stress, which is compensated by unhealthy food consumption.

Stress is a negative factor that affects health, so food consumption as a means to cope with stress, from which individuals derive hedonistic pleasure whether the food is sweet or not, provides comfort and alleviates the reaction to stress [35]. If the consumption of comfort food helps a person recover faster after a stressful event, then such behavior can be protective. Namely, studies have confirmed that sugar or a sweet taste can relieve stress by inhibiting cortisol secretion or activating endogenous cannabinoid receptors [35,44]. The intake of comfort food, usually unhealthy and high in fat or sugar, dampens the stress-induced HPA responses of adrenocorticotropic hormone and cortisol secretion [45]. However, immediate relief of the pathophysiological mechanisms of stress is not an end in itself because poor recovery from stress negatively affects the burden on the body and contributes to poor health outcomes in the long term. Studies have confirmed that people with psychophysiological stress are at risk of consuming unhealthy comfort foods [28]. Furthermore, unhealthy eating actually has a comfort connection and confirms the association between emotions and lifestyle with a negative effect on physical health [28]. In a pathophysiological sense, unhealthy comfort eating reduces stress-induced activation of the hypothalamic–pituitary–adrenal axis (HPA) [45]. Additionally, this mechanism of dampening stress reactivity fosters abdominal adiposity, which results in chronically elevated glucocorticoid levels and helps to inhibit HPA axis reactivity in those moments when it may be needed [28,35,46]. In the long term, these ways of stress relief have a negative effect not only due to the suppression and impairment of the HPA axis but also on the different pathophysiological mechanisms and encouragement of oxidative stress [47]. Moreover, stress is associated with the neurotransmitters and hormones that control appetite and is positively associated with a higher-fat diet [23]. Under conditions of greater perceived stress, increases in total energy and fat intake have been reported [23]. All of these described mechanisms associated with the overconsumption of sweet, salty, and fatty foods may increase the risk of a higher incidence of chronic diseases, such as cardiovascular diseases, hypertension, hypercholesterolemia, diabetes, obesity, gastrointestinal disorders, cancer, and other noncommunicable chronic diseases [11,12,13,14,48].

Various studies have suggested that a high fat or sugar content is not necessary for food to be pleasurable and stress-protective [35]. Although it is assumed that in these states, the consumption of food is abundant, especially foods rich in energy, fat, and sugar, the question still arises as to whether all food consumed in the mentioned states is unhealthy [35]. For example, fruits and vegetables also contain considerable amounts of sugar, and thus, in addition to relieving stress, they can have positive effects due to their content of dietary fiber and other antioxidant substances and can have a strong preventive effect. Awareness of the significance of dietary fiber within dietary patterns allows consumers to choose healthy food in lieu of unhealthy food [49,50,51]. Hyperpalatable foods, which are typically ultra-processed and abundant in high energy density, excess sugar, fat, and salt, as well as meat and animal products, need to be substituted with fruits and vegetables, such as legumes, in order to prevent detrimental impact on health [51,52,53]. This food is nutritious, high in protein, and nutrient-dense, which can avoid a number of chronic diseases, such as heart disease, stroke, diabetes, bowel cancer, and obesity [53].

Despite this, stress may contribute to listlessness, social isolation, fatigue, and less physical activity. In addition to downregulating the cognitive control centers of the brain, stress activates “reward pathways” and increases cravings for palatable foods. This combination of neural adaptations frequently leads to an increase in the consumption of palatable foods, such as comfort foods high in fat and/or sugar [45]. Additionally, the typical adaptive response to negative emotion is loss of appetite [34]. Stress can especially affect the appetite through hormonal regulation. For example, in acute stress, noradrenaline suppresses the appetite, while in chronic stress, cortisol stimulates the appetite [23]. Additionally, regularly eating energy-dense food without hunger may result in higher weight gain and obesity under stress conditions [34]. In our study, health and agriculture professionals, people who never exercise, and people who are overweight are at risk of consuming food when they are under stress. This can be explained by stress in the workplace, which is closely related to the health profession. Furthermore, since physical activity can significantly contribute to stress relief and the experience of stress impairs the efforts of individuals to be physically active, it is possible that people who do not exercise perceive a higher level of stress [54].

Apart from stress, the relationship between emotional reasons or conditions such as a depressive mood, loneliness, boredom, and emotional consolation as reasons for food consumption and the predictors of food consumption, such as country of residence, age, gender, marital status, profession, physical activity, adherence to a healthy diet, motivation for health behavior, and emotional motivation, confirm the complexity of the factors that are reflected in these conditions. Namely, due to different cultural backgrounds, well-being, and especially the wealth of a society, such as the GDP per capita of the countries, differences in these motivational factors for food consumption may be expected [38,55,56]. Moreover, in comparison to Croatia, almost all of the other European countries had a higher probability of seeking consolation via eating when feeling lonely and bored, while the Croatian respondents more than those from the other countries use food as emotional consolation in general.

Social relationships are part of an individual’s environment. When an individual perceives a deficiency in social relationships, they may feel lonely [57]. Moreover, social isolation and loneliness are associated with physical and mental comorbidity and increased mortality [58]. The lower risk of seeking consolation through eating when lonely and bored and the higher risk of emotional consolation in general in the Croatian respondents may be explained by cultural or social characteristics. Nevertheless, some studies have noted that it is yet unclear whether cultural background affects the development of loneliness [57]. Despite this, studies have confirmed a link between loneliness and eating disorders [59].

Our results confirmed a greater risk of eating sweets in depressive moods in rural respondents, female respondents, and those with higher emotional motivation. The risk of eating sweets when individuals feel depressed in rural areas may be explained by the results of several studies that showed that those living in rural environments experience worse health outcomes, such as depression, diabetes, heart disease, stroke, cancer, and suicide [60]. Other studies have also confirmed that depression is mediated by emotional eating and may affect BMI [12]. Some studies have confirmed that women in rural areas are at risk of depressive symptoms, loneliness, and social isolation due to possible greater distances from friends and family, lower educational achievements, and unavailability of social and health services, and there may be differences in health status [61,62]. Additionally, when compared to the Croatian respondents, the Hungarian, Slovenian, Greek, and Italian respondents ate more sweets when depressed, while the Latvian and Portuguese respondents ate sweets less often. It should be pointed out that the percentage of rural respondents was slightly higher in the Hungarian and Slovenian populations; almost one-third of the respondents were women for both of these countries [62,63]. Additionally, previous studies have noted that women are more likely than men to report a depressive mood, which may be associated with our results in that women are at risk of eating sweets when depressed [62,63].

The unemployed, female, and rural respondents and health, sport, and food professionals had a lower probability of consuming food (e.g., wine or tea) for the purpose of relaxation, which may indicate that these groups are under higher stress and rarely use these types of relaxation that have certain health advantages. For example, a daily intake of red wine of 200 mL with meals is associated with the prevention of cardiovascular diseases and breast cancer and contributes to relaxation, which is attributed to the phytochemicals and not to the alcohol content [64,65]. Additionally, studies have shown that moderate wine consumption is associated with positive emotions, and motivation for consumption is related to context, e.g., special occasions [66]. Additionally, the Portuguese respondents had the highest probability of consuming this type of food when compared to the Croatian respondents, which can be explained by previous studies that have confirmed that Portugal is considered a wine-drinking country [66].

Compared to the Croatian respondents, the Lithuanian, Serbian, Latvian, Dutch, and Italian respondents were less likely to use food to regulate their body weight. It is possible that the Mediterranean environment contributes to this because most of these countries are not in the Mediterranean except Italy. However, previous studies among Croatian participants have shown that the consumption of a Mediterranean diet in the Croatian population is quite low [67,68]. In contrast, the Slovenian and Portuguese respondents, who also live in the Mediterranean region, have a higher probability of consuming food that helps control weight compared to the Croatian respondents. The reasons can be found in other factors. For example, the Slovenian and Portuguese respondents conducted more exercise activity than the other respondents, while the Lithuanian and Polish respondents had higher MHB and normal BMI. It is possible that MHB contributes to the lifestyles of these study groups. Furthermore, it was observed that those who live in rural environments and live a single life have a higher risk of not consuming foods that help them control their body weight. This can be explained by social relationships because it has been confirmed that the family environment often contributes to more frequent meal preparation rather than the consumption of fast food, which favors obesity. Moreover, studies have shown that not exercising contributes to obesity and bad mood, which was confirmed in our research, in which respondents had a lower probability of food consumption for better relaxation, feeling good, or controlling weight. Furthermore, in our study, greater health motivation and emotional motivation were positively reflected in the consumption of food that preserves normal body weight and good feelings. This may mean that the subjects react emotionally to possible weight gain, which may also indicate that emotional motivation does not necessarily include the consumption of harmful foods [35].

On the contrary, those respondents with higher emotional motivation had a higher probability of consuming food to keep themselves awake and alert, while those respondents with higher motivation for health behavior had a lower probability of engaging in this harmful behavior. Additionally, it has been observed that health workers consume food more often to keep awake and alert. It is understandable that the nature of their work requires constant vigilance and concentration, and in addition, there is constant exposure to stress. It is possible that as a result of the above, these individuals show a greater need for consuming more of these foods [69,70]. Nevertheless, previous studies have been unclear on how caffeine affects professionals’ clinical performance [70].

### Limitations

This study analyzed an important segment of lifestyle both because of the importance of the topic it deals with in the field of lifestyle medicine and because of different psychological, cultural, and social differences in a large number of European countries that were included in this study. Despite this, this study has some limitations. First, this was a cross-sectional study; thus, it cannot contribute to the discovery of causality associations between EEB and unhealthy lifestyle habits, demographic characteristics, motivation for healthy behavior, and emotional conditions. Second, we did not examine what our subjects ate in the mentioned states of stress, depression, or loneliness (e.g., sweets, fatty foods, snacks, or maybe some healthy foods). Third, we used the self-assessment method to assess weight and height, which may have affected the objectivity of the data. Fourth, our questionnaire was conducted to assess food choice determinants according to types of conditioning motivation, not only emotional eating. Finally, we did not assess how common this pattern of EEB was in our study population, which could contribute to areas for future research.

Nevertheless, this study provides a valuable scientific contribution, not only because of its important features in areas of nutrition by linking the significance of the psychological impact of certain psychological states on food choice and consumption but also because of the emphasis on their importance for health outcomes. Further research is necessary to clarify the associations between the reasons for food consumption, coping with stress, emotions, and emotional food consolation in order to prevent new chronic diseases [4].

## 5. Conclusions

This study included a respectable number of European countries as a model of assessment of emotional eating behavior, adding a crucial contribution to this demanding field in lifestyle medicine. This study confirmed the associations between different emotional conditions and food consumption. The results showed that emotions might provoke emotional eating behavior, which may contribute to the risk of developing chronic diseases, such as obesity, cardiovascular disease, and other adverse health consequences. It is necessary to make the population aware of ways to deal with emotional states. In states overwhelmed by emotions, it is necessary to think about healthy ways of dealing with stress, depression, or other emotional states. Instead of emotional eating behavior and consumption of unhealthy food, emphasis should be placed on healthy lifestyle habits, such as physical exercise and healthy eating habits. Thereby, public health programs are urgently needed to stop unhealthy dietary patterns and unhealthy lifestyles and prevent adverse health consequences on human health.

## Figures and Tables

**Figure 1 foods-12-00872-f001:**
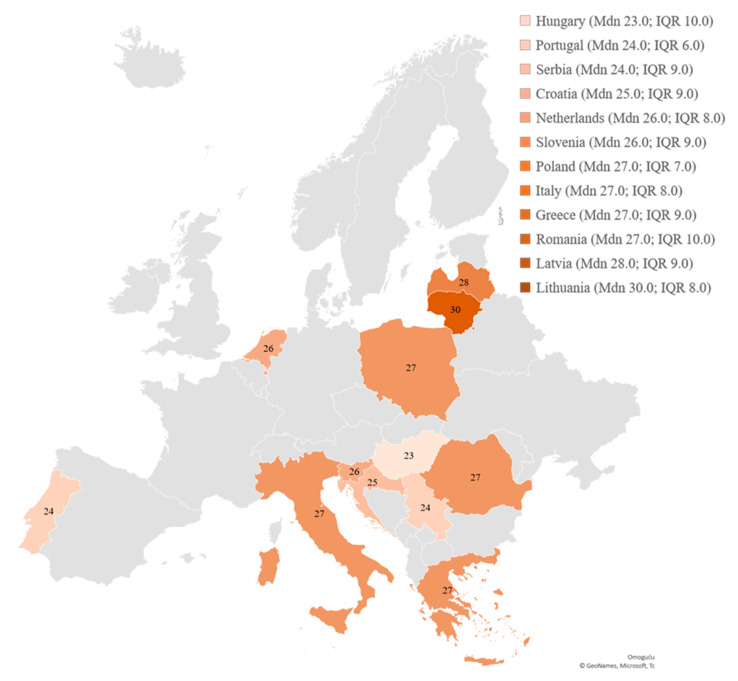
Overall emotional eating behavior among the European countries by median values (ranging from 9 to 45); Countries’ colors ranged from lighter to darker; lighter colors indicated a lower frequency, and darker ones a greater frequency of food consumption in various emotional conditions.

**Table 1 foods-12-00872-t001:** Sociodemographic data, lifestyle characteristics, and motivation for health behavior of the study population (*N* = 9052).

		Croatia (*N* = 1538)	Greece (*N* = 498)	Hungary (*N* = 500)	Italy (*N* = 541)	Latvia (*N* = 636)	Lithuania (*N* = 507)	The Netherlands (*N* = 521)	Poland (*N* = 586)	Portugal (*N* = 1314)	Romania (*N* = 821)	Serbia (*N* = 498)	Slovenia (*N* = 1093)	Overall *p*
Age (years), Mdn (IQR)		32.0 (25.0)	23.0 (22.0)	43.0 (18.0)	40.0 (19.0)	36.0 (19.0)	25.0 (24.0)	29.0 (25.0)	30.0 (18.0)	37.0 (26.0)	38.0 (24.0)	23.0 (7.0)	30.0 (17.0)	<0.001 *
Gender, *N* (%)	Female	1052 (68.4)	317 (63.7)	253 (50.6)	354 (65.4)	518 (81.4)	382 (75.3)	451 (86.6)	447 (76.3)	881 (67.0)	566 (68.9)	323 (64.9)	955 (87.5)	<0.001 ^†^
	Male	486 (31.6)	181 (36.3)	247 (49.4)	187 (34.6)	118 (18.6)	125 (24.7)	70 (13.4)	139 (23.7)	433 (33.0)	255 (31.1)	175 (35.1)	137 (12.5)	
Education, *N* (%)	Primary and high school	716 (46.6)	67 (13.5)	339 (67.8)	271 (50.1)	146 (23.0)	229 (45.2)	113 (21.7)	111 (18.9)	570 (43.4)	182 (22.2)	151 (30.3)	535 (49.0)	<0.001 ^†^
	University	822 (53.4)	431 (86.5)	161 (32.2)	270 (49.9)	490 (77.0)	278 (54.8)	408 (78.3)	475 (81.1)	744 (56.6)	639 (77.8)	347 (69.7)	557 (51.0)	
Residential environment, *N* (%)	Urban	1318 (85.7)	474 (95.2)	358 (71.6)	481 (88.9)	544 (85.5)	457 (90.1)	497 (95.4)	516 (88.1)	1099 (83.6)	707 (86.1)	466 (93.6)	747 (68.4)	<0.001 ^†^
	Rural	220 (14.3)	24 (4.8)	142 (28.4)	60 (11.1)	92 (14.5)	50 (9.9)	24 (4.6)	70 (11.9)	215 (16.4)	114 (13.9)	32 (6.4)	345 (31.6)	
Marital status, *N* (%)	Married	724 (47.1)	154 (30.9)	324 (64.8)	357 (66.0)	361 (56.8)	242 (47.7)	257 (49.3)	386 (65.9)	654 (49.8)	524 (63.8)	108 (21.7)	689 (63.1)	<0.001 ^†^
	Single/divorced/widowed	814 (52.9)	344 (69.1)	176 (35.2)	184 (34.0)	275 (43.2)	265 (52.3)	264 (50.7)	200 (34.1)	660 (50.2)	297 (36.2)	390 (78.3)	403 (36.9)	
Employment status, *N* (%)	Employed	998 (64.9)	214 (43)	388 (77.6)	425 (78.6)	538 (84.6)	248 (48.9)	287 (55.1)	440 (75.1)	860 (65.4)	591 (72)	147 (29.5)	778 (71.2)	<0.001 ^†^
	Unemployed/retired/student	540 (35.1)	284 (57)	112 (22.4)	116 (21.4)	98 (15.4)	259 (51.1)	234 (44.9)	146 (24.9)	454 (34.6)	230 (28)	351 (70.5)	314 (28.8)	
Profession, *N* (%)	Nutrition	74 (4.8)	29 (5.8)	7 (1.4)	26 (4.8)	14 (2.2)	33 (6.5)	58 (11.1)	69 (11.8)	36 (2.7)	108 (13.2)	30 (6)	191 (17.5)	<0.001 ^†^
	Food	148 (9.6)	106 (21.3)	38 (7.6)	58 (10.7)	182 (28.6)	161 (31.8)	65 (12.5)	141 (24.1)	90 (6.8)	29 (3.5)	58 (11.6)	34 (3.1)	
	Agriculture	67 (4.4)	22 (4.4)	16 (3.2)	14 (2.6)	37 (5.8)	17 (3.4)	33 (6.3)	51 (8.7)	42 (3.2)	12 (1.5)	50 (10)	10 (0.9)	
	Sport	33 (2.1)	8 (1.6)	14 (2.8)	21 (3.9)	22 (3.5)	17 (3.4)	3 (0.6)	26 (4.4)	54 (4.1)	21 (2.6)	8 (1.6)	57 (5.2)	
	Psychology	31 (2.0)	4 (0.8)	17 (3.4)	18 (3.3)	19 (3.0)	9 (1.8)	38 (7.3)	15 (2.6)	29 (2.2)	30 (3.7)	19 (3.8)	31 (2.8)	
	Health	350 (22.8)	47 (9.4)	34 (6.8)	55 (10.2)	192 (30.2)	82 (16.2)	81 (15.5)	6 (1.0)	187 (14.2)	322 (39.2)	59 (11.8)	152 (13.9)	
	Other	835 (54.3)	282 (56.6)	374 (74.8)	349 (64.5)	170 (26.7)	188 (37.1)	243 (46.6)	278 (47.4)	876 (66.7)	299 (36.4)	274 (55)	617 (56.5)	
Following a healthy diet, Mdn (IQR)		3.0 (1.0)	4.0 (1.0)	3.0 (1.0)	4.0 (1.0)	3.0 (1.0)	3.0 (1.0)	4.0 (1.0)	3.0 (1.0)	3.0 (1.0)	3.0 (1.0)	3.0 (1.0)	4.0 (0.0)	<0.001 *
Body mass index, Mdn (IQR)		23.5 (5.4)	22.9 (4.7)	25.9 (7.1)	23.1 (5.0)	24.3 (5.3)	23.5 (6.1)	23.3 (5.3)	23.1 (5.3)	21.5 (3.2)	24.2 (5.6)	22.3 (4.7)	23.6 (5.3)	<0.001 *
Category of body mass index, *N* (%)	Underweight (≤18.5)	52 (3.4)	14 (2.8)	13 (2.6)	32 (5.9)	15 (2.4)	27 (5.3)	19 (3.6)	26 (4.5)	72 (5.5)	42 (5.1)	36 (7.2)	40 (3.7)	<0.001 ^†^
	Normal (18.5–24.9)	922 (59.9)	340 (68.3)	201 (40.2)	342 (63.2)	354 (55.8)	285 (56.2)	325 (62.4)	371 (63.6)	1042 (79.9)	442 (53.8)	344 (69.1)	659 (60.4)	
	Overweight (25.0–29.9)	463 (30.1)	117 (23.5)	166 (33.2)	130 (24)	176 (27.8)	144 (28.4)	113 (21.7)	133 (22.8)	148 (11.3)	242 (29.5)	96 (19.3)	270 (24.7)	
	Obesity (≥30.0)	101 (6.6)	27 (5.4)	120 (24.0)	37 (6.8)	89 (14.0)	51 (10.1)	64 (12.3)	53 (9.1)	42 (3.2)	95 (11.6)	22 (4.4)	122 (11.2)	
Exercise activity, *N* (%)	No	598 (38.9)	176 (35.3)	263 (52.6)	221 (40.9)	201 (31.6)	286 (56.4)	177 (34)	217 (37)	297 (22.6)	324 (39.5)	167 (33.5)	198 (18.1)	<0.001 ^†^
	Yes	940 (61.1)	322 (64.7)	237 (47.4)	320 (59.1)	435 (68.4)	221 (43.6)	344 (66)	369 (63)	1017 (77.4)	497 (60.5)	331 (66.5)	894 (81.9)	
Television or computer (hours/day), Mdn (IQR)		3.0 (4.0)	3.0 (3.0)	3.0 (3.0)	4.0 (6.0)	6.0 (5.0)	3.0 (3.0)	4.0 (3.3)	4.0 (6.0)	6.0 (5.0)	4.0 (5.0)	3.0 (3.0)	2.0 (3.0)	<0.001 *
Motivation for health behavior, Mdn (IQR)		34.0 (7.0)	34.0 (8.0)	31.0 (7.0)	34.0 (6.0)	34.0 (6.8)	36.0 (7.0)	30.0 (7.0)	35.0 (6.0)	38.0 (7.0)	30.0 (8.0)	33.0 (7.0)	35.0 (6.0)	<0.001 *

Note: *Mdn (IQR)* = median (interquartile range); *N* (%) = absolute number (percentage); other profession = profession not related to the list; heath motivation range = 10–50. * Kruskal–Wallis test; ^†^ chi-square test.

**Table 2 foods-12-00872-t002:** The difference in emotional conditions of food consumption of the study population (*N* = 9052).

		Croatia (*N* = 1538)	Greece (*N* = 498)	Hungary (*N* = 500)	Italy (*N* = 541)	Latvia (*N* = 636)	Lithuania (*N* = 507)	The Netherlands (*N* = 521)	Poland (*N* = 586)	Portugal (*N* = 1314)	Romania (*N* = 821)	Serbia (*N* = 498)	Slovenia (*N* = 1093)	Overall *p* *
Coping with stress	Mdn (IQR)	3.0 (2.0)	3.0 (2.0)	2.0 (2.0)	3.0 (2.0)	3.0 (2.0)	3.0 (2.0)	3.0 (2.0)	3.0 (2.0)	3.0 (2.0)	3.0 (2.0)	3.0 (2.0)	3.0 (2.0)	<0.001
	Mean Rank	4266.4	4792.9	3324.5	4924.7	5042.0	5399.1	4741.8	4747.8	3935.6	4688.1	4318.3	4883.0	
Eating sweets when depressed	Mdn (IQR)	3.0 (2.0)	3.0 (2.0)	3.0 (3.0)	3.0 (2.0)	3.0 (2.0)	3.0 (2.0)	3.0 (2.0)	3.0 (2.0)	2.0 (1.0)	3.0 (2.0)	2.0 (3.0)	3.0 (2.0)	<0.001
	Mean Rank	4356.3	4756.1	4169.1	4915.2	4847.6	5308.9	4824.6	4686.7	3781.2	4742.6	3966.4	4844.9	
Consolation via eating when lonely	Mdn (IQR)	2.0 (2.0)	2.0 (1.0)	2.0 (2.0)	2.0 (1.0)	3.0 (1.0)	3.0 (2.0)	3.0 (2.0)	2.0 (1.0)	2.0 (0.0)	2.0 (3.0)	2.0 (2.0)	2.0 (2.0)	<0.001
	Mean Rank	4105.3	4800.5	3729.1	4820.0	5424.2	5779.5	5106.7	4498.6	4042.6	4783.6	3857.3	4543.3	
Boredom eating	Mdn (IQR)	3.0 (2.0)	3.0 (2.0)	2.0 (2.0)	3.0 (2.0)	3.0 (2.0)	4.0 (1.0)	4.0 (2.0)	4.0 (2.0)	2.0 (2.0)	3.0 (2.0)	3.0 (2.0)	3.0 (2.0)	<0.001
	Mean Rank	4368.8	4891.9	3459.4	4940.5	5340.2	5523.5	5298.2	4923.9	3687.2	4511.2	4375.4	4438.5	
Emotional consolation	Mdn (IQR)	3.0 (1.0)	2.0 (1.0)	2.0 (2.0)	2.0 (1.0)	3.0 (2.0)	3.0 (2.0)	3.0 (1.0)	2.0 (2.0)	2.0 (1.0)	2.0 (2.0)	2.0 (1.0)	2.0 (2.0)	<0.001
	Mean Rank	4756.4	4541.5	3936.1	4683.7	5631.0	5916.6	4789.1	4691.1	4011.9	4212.5	3373.8	4268.1	
Helping to control weight	Mdn (IQR)	3.0 (2.0)	3.0 (2.0)	2.0 (2.0)	3.0 (2.0)	3.0 (2.0)	3.0 (2.0)	3.0 (2.0)	3.0 (4.0)	4.0 (1.0)	3.0 (1.0)	2.0 (2.0)	4.0 (1.0)	<0.001
	Mean Rank	4378.6	4327.8	3680.4	4179.1	4062.6	4031.5	3597.8	4847.2	5714.0	4898.7	3278.6	5017.2	
Keeping awake and alert	Mdn (IQR)	3.0 (2.0)	2.0 (3.0)	3.0 (2.0)	2.0 (2.0)	3.0 (1.0)	3.0 (2.0)	2.0 (2.0)	3.0 (2.0)	2.0 (2.0)	3.0 (2.0)	3.0 (2.0)	2.0 (3.0)	<0.001
	Mean Rank	4595.8	4230.0	5144.8	4443.9	5520.0	5148.1	4065.1	4547.4	3362.0	5416.5	5040.4	4162.4	
Helping to relax	Mdn (IQR)	3.0 (2.0)	3.0 (2.0)	3.0 (1.0)	3.0 (2.0)	3.0 (2.0)	3.0 (2.0)	2.0 (2.0)	3.0 (1.0)	4.0 (1.0)	3.0 (1.0)	3.0 (2.0)	3.0 (2.0)	<0.001
	Mean Rank	4213.3	4523.5	3619.6	4476.2	4776.2	4838.8	3599.8	5040.6	5684.3	4613.7	4681.0	3756.9	
Making oneself feel good	Mdn (IQR)	4.0 (1.0)	4.0 (2.0)	4.0 (1.0)	3.0 (2.0)	4.0 (2.0)	4.0 (0.0)	4.0 (1.0)	4.0 (1.0)	4.0 (0.0)	4.0 (1.0)	4.0 (2.0)	4.0 (1.0)	<0.001
	Mean Rank	4374.2	4879.4	3831.4	4885.2	5575.1	5158.0	4324.6	4625.0	4913.1	4012.2	3780.2	4122.9	
Emotional eating behavior (overall)	Mdn (IQR)	25.0 (9.0)	27.0 (9.0)	23.0 (10.0)	27.0 (8.0)	28.0 (9.0)	30.0 (8.0)	26.0 (8.0)	27.0 (7.0)	24.0 (6.0)	27.0 (10.0)	24.0 (9.0)	26.0 (9.0)	<0.001
	Mean Rank	4301.70	4744.37	3453.4	4835.13	5549.24	5800.0	4503.88	4915.59	4062.89	4789.99	3757.53	4407.69	

Note: *Mdn (IQR)* = median (interquartile range); *Mean rank* = average of the ranks for all observations within each sample; emotional eating behavior (overall) range = 9–45. * Kruskal–Wallis test.

**Table 3 foods-12-00872-t003:** Associations between the emotional conditions of food consumption and the respondents’ characteristics using an ordinal regression model, *N* = 9052.

	Coping with Stress	Eating Sweets when Depressed	Consolation via Eating when Lonely	Boredom Eating	Emotional Consolation
	OR * (95% CI ^†^); *p* ^‡^	OR * (95% CI ^†^); *p* ^‡^	OR * (95% CI ^†^); *p* ^‡^	OR * (95% CI ^†^); *p* ^‡^	OR * (95% CI ^†^); *p* ^‡^
Country of residence (reference = Croatia)					
Greece	1.30 (1.07–1.60); 0.010	1.28 (1.05–1.57); 0.015	1.90 (1.53–2.35); <0.001	1.35 (1.10–1.65); 0.004	0.55 (0.44–0.68); <0.001
Hungary	0.56 (0.45–0.69); <0.001	1.64 (1.34–2.02); <0.001	1.02 (0.81–1.28); 0.879	0.61 (0.50–0.75); <0.001	0.63 (0.51–0.79); <0.001
Italy	1.40 (1.15–1.69); 0.001	1.27 (1.05–1.54); 0.015	1.42 (1.15–1.74); 0.001	1.50 (1.24–1.82); <0.001	0.55 (0.45–0.68); <0.001
Latvia	0.83 (0.69–1.00); 0.049	0.56 (0.46–0.67); <0.001	1.40 (1.15–1.70); 0.001	1.38 (1.14–1.66); 0.001	0.68 (0.56–0.82); <0.001
Lithuania	1.10 (0.90–1.35); 0.347	0.87 (0.71–1.07); 0.183	2.33 (1.88–2.88); <0.001	1.33 (1.09–1.62); 0.006	0.98 (0.79–1.20); 0.819
Netherlands	1.41 (1.15–1.72); 0.001	1.14 (0.94–1.39); 0.187	2.74 (2.22–3.38); <0.001	2.02 (1.66–2.46); <0.001	0.70 (0.57–0.86); 0.001
Poland	1.10 (0.91–1.33); 0.330	0.88 (0.73–1.06); 0.180	1.16 (0.94–1.42); 0.161	1.31 (1.09–1.59); 0.005	0.57 (0.47–0.69); <0.001
Portugal	1.08 (0.93–1.25); 0.329	0.77 (0.67–0.90); 0.001	1.66 (1.42–1.96); <0.001	0.81 (0.70–0.94); 0.006	0.63 (0.54–0.73); <0.001
Romania	1.08 (0.91–1.28); 0.369	1.02 (0.86–1.20); 0.856	1.68 (1.40–2.02); <0.001	0.93 (0.79–1.10); 0.406	0.28 (0.23–0.33); <0.001
Serbia	1.62 (1.32–1.98); <0.001	1.01 (0.82–1.23); 0.947	1.36 (1.09–1.70); 0.006	1.26 (1.03–1.54); 0.022	0.28 (0.22–0.34); <0.001
Slovenia	1.90 (1.62–2.23); <0.001	1.45 (1.24–1.70); <0.001	1.94 (1.64–2.30); <0.001	1.03 (0.88–1.20); 0.750	0.56 (0.47–0.66); <0.001
Age (years; reference = elderly (≥66 years))					
Young adults (18–30 years)	1.02 (0.78–1.34); 0.882	0.74 (0.57–0.97) 0.030	0.49 (0.36–0.65); <0.001	2.11 (1.61–2.77); <0.001	0.62 (0.47–0.82); 0.001
Middle-aged adults (31–50 years)	1.11 (0.84–1.46); 0.461	0.89 (0.68–1.18) 0.425	0.72 (0.54–0.96); 0.026	1.66 (1.26–2.18); <0.001	0.81 (0.61–1.07); 0.136
Senior adults (51–65 years)	1.16 (0.88–1.53); 0.304	0.93 (0.70–1.22) 0.591	0.86 (0.64–1.16); 0.321	1.54 (1.16–2.04); 0.002	0.95 (0.71–1.28); 0.755
Gender (reference = male)					
Female	1.04 (0.94–1.15); 0.417	2.08 (1.89–2.30); <0.001	1.14 (1.03–1.26); 0.014	1.19 (1.08–1.31); <0.001	1.22 (1.11–1.35) <0.001
Education (reference = university)					
No university	0.99 (0.90–1.09); 0.843	0.90 (0.82–0.99); 0.035	1.13 (1.02–1.24); 0.019	1.02 (0.93–1.12); 0.720	0.89 (0.81–0.99); 0.024
Residential environment (reference = urban)					
Rural	1.09 (0.97–1.23); 0.138	1.19 (1.06–1.34); 0.003	0.96 (0.85–1.08); 0.490	1.17 (1.04–1.32); 0.008	0.96 (0.85–1.08); 0.469
Marital status (reference = married)					
Single, divorced, and widowed	1.06 (0.97–1.17); 0.216	1.08 (0.98–1.19); 0.106	1.14 (1.02–1.26); 0.015	0.98 (0.89–1.08); 0.638	0.95 (0.86–1.05); 0.285
Employment (reference = employed)					
Unemployed	1.02 (0.92–1.14); 0.656	1.03 (0.92–1.14); 0.613	1.05 (0.93–1.17) 0.437	1.05 (0.95–1.17); 0.356	1.14 (1.02–1.27); 0.025
Profession (reference = other profession ^§^)					
Nutrition	1.07 (0.91–1.26); 0.406	1.06(0.90–1.25); 0.477	0.82 (0.69–0.98); 0.026	0.92 (0.78–1.08); 0.295	1.01 (0.85–1.20); 0.891
Food	1.10 (0.96–1.25); 0.180	1.06 (0.93–1.21); 0.384	0.88 (0.77–1.02); 0.088	0.82 (0.72–0.94); 0.004	1.18 (1.03–1.36); 0.019
Agriculture	1.30 (1.06–1.61); 0.013	0.98 (0.80–1.21); 0.879	0.80 (0.64–1.00); 0.049	0.83 (0.67–1.02); 0.071	0.79 (0.63–0.98); 0.036
Sport	1.13 (0.90–1.44); 0.296	1.14 (0.90–1.45); 0.263	0.91 (0.71–1.17); 0.470	0.86 (0.68–1.08); 0.191	1.07 (0.84–1.37); 0.590
Psychology	1.26 (0.98–1.61); 0.069	1.14 (0.90–1.46); 0.283	0.94 (0.72–1.22); 0.627	0.87 (0.68–1.11); 0.273	0.87 (0.67–1.12); 0.270
Health	1.15 (1.02–1.29); 0.020	0.95 (0.85–1.07); 0.423	0.94 (0.83–1.07); 0.339	0.92 (0.82–1.03); 0.144	1.35 (1.20–1.53) <0.001
Following a healthy diet (reference = yes)					
Never or rarely follow	1.22 (1.08–1.38); 0.001	1.12 (0.99–1.27); 0.064	1.16 (1.02–1.32); 0.022	0.93 (0.82–1.05); 0.247	0.99 (0.87–1.12); 0.817
Body mass index (reference = obesity (BMI ^║^ ≥ 30.0))					
Underweight (BMI < 18.5)	0.65 (0.50–0.83); 0.001	0.90 (0.71–1.16); 0.426	0.56 (0.43–0.74); <0.001	0.64 (0.50–0.82); <0.001	0.60 (0.46–0.77); <0.001
Normal weight (18.5 ≤ BMI ≤ 24.9)	0.97 (0.96–0.98); <0.001	0.96 (0.83–1.12); 0.635	0.57 (0.48–0.67); <0.001	0.82 (0.70–0.95); 0.009	0.61 (0.52–0.71); <0.001
Overweight (25.0 ≤ BMI ≤ 29.9)	1.42 (1.41–1.44); <0.001	1.01 (0.87–1.18); 0.881	0.71 (0.61–0.84); <0.001	0.88 (0.75–1.02); 0.097	0.73 (0.62–0.86); <0.001
Physical exercise (reference = weekly)					
Never exercise	0.96 (0.87–1.05); 0.331	1.07 (0.98–1.17); 0.122	1.15 (1.04–1.27); 0.004	1.25 (1.14–1.37); <0.001	1.15 (1.05–1.26); 0.004
Sitting in front of a television or computer (hours per day)	0.99 (0.98–1.01); 0.195	0.98 (0.96–0.99); 0.004	0.99 (0.97–1.00); 0.158	0.99 (0.97–1.00); 0.077	0.99 (0.98–1.01); 0.239
Motivation for health behavior (overall sum)	1.08 (0.93–1.25); 0.329	0.97 (0.96–0.98); <0.001	0.94 (0.93–0.95); <0.001	0.94 (0.93–0.95); <0.001	0.95 (0.94–0.95); <0.001
Emotional eating behavior (overall sum)	1.30 (1.07–1.60); 0.010	1.41 (1.40–1.43); <0.001	1.60 (1.58–1.62); <0.001	1.37 (1.36–1.39); <0.001	1.55 (1.54–1.57); <0.001

Note: * OR = adjusted odds ratio; ^†^ CI = 95% confidence interval; ^‡^
*p* = *p*-value; ^§^ Other profession = professions not related to the list; ^║^ BMI = body mass index.

**Table 4 foods-12-00872-t004:** Associations between the respondents’ characteristics and food consumption as a way of improving physical and psychological conditions using an ordinal regression model, *N* = 9052.

	Helping to Control Weight	Keeping Awake and Alert	Relaxation	Making Oneself Feel Good
	OR * (95% CI ^†^); *p* ^‡^	OR * (95% CI ^†^); *p* ^‡^	OR * (95% CI ^†^); *p* ^‡^	OR * (95% CI ^†^); *p* ^‡^
Country of residence (reference = Croatia)				
Greece	0.87 (0.71–1.06); 0.159	0.62 (0.51–0.75); <0.001	1.04 (0.86–1.26); 0.664	1.21 (0.99–1.49); 0.060
Hungary	1.12 (0.92–1.36); 0.279	1.84 (1.52–2.23); <0.001	0.99 (0.82–1.21); 0.955	1.04 (0.85–1.27); 0.689
Italy	0.71 (0.59–0.86); <0.001	0.72 (0.60–0.87); <0.001	0.98 (0.82–1.18); 0.866	1.23 (1.01–1.49); 0.039
Latvia	0.65 (0.54–0.78); <0.001	1.21 (1.01–1.44); 0.037	1.15 (0.96–1.37); 0.133	1.95 (1.61–2.36); <0.001
Lithuania	0.35 (0.29–0.43); <0.001	1.00 (0.83–1.21); 0.991	0.96 (0.79–1.16); 0.673	0.89 (0.73–1.09); 0.280
Netherlands	0.79 (0.65–0.96); 0.015	0.59 (0.49–0.71); <0.001	0.65 (0.54–0.79); <0.001	0.98 (0.80–1.19); 0.830
Poland	1.15 (0.95–1.39); 0.140	0.83 (0.69–0.99); 0.044	1.47 (1.22–1.76); <0.001	0.80 (0.66–0.97); 0.020
Portugal	1.53 (1.32–1.78); <0.001	0.43 (0.38–0.50); <0.001	2.86 (2.47–3.30); <0.001	1.36 (1.17–1.58); <0.001
Romania	0.99 (0.84–1.17); 0.916	1.96 (1.67–2.30); <0.001	1.11 (0.95–1.31); 0.192	0.51 (0.43–0.61); <0.001
Slovenia	1.29 (1.10–1.50); 0.001	0.80 (0.69–0.93); 0.004	0.65 (0.56–0.76); <0.001	0.70 (0.60–0.82); <0.001
Serbia	0.49 (0.40–0.60); <0.001	1.73 (1.43–2.09); <0.001	1.58 (1.31–1.92); <0.001	0.64 (0.52–0.77); <0.001
Age (years; reference = elderly (≥66 years))				
Young adults (18–30 years)	0.81 (0.62–1.05); 0.110	1.69 (1.30–2.20); <0.001	0.89 (0.69–1.15); 0.368	0.91 (0.70–1.20); 0.508
Middle-aged adults (31–50 years)	0.87 (0.66–1.13); 0.294	1.69 (1.30–2.21); <0.001	0.72 (0.56–0.94); 0.015	0.71 (0.54–0.94); 0.015
Senior adults (51–65 years)	0.84 (0.64–1.11); 0.222	1.30 (0.99–1.71); 0.055	0.78 (0.60–1.02); 0.073	0.78 (0.59–1.03); 0.084
Gender (reference = male)				
Female	1.03 (0.94–1.14); 0.491	0.63 (0.57–0.69); <0.001	0.81 (0.74–0.89); <0.001	0.70 (0.63–0.77); <0.001
Education (reference = university)				
No university	0.93 (0.85–1.01); 0.099	1.09 (1.00–1.19); 0.055	1.04 (0.96–1.14); 0.332	0.94 (0.86–1.04); 0.226
Residential environment (reference = urban)				
Rural	0.88 (0.79–0.99); 0.027	0.97 (0.87–1.08); 0.563	0.85 (0.76–0.95); 0.005	0.91 (0.81–1.02); 0.109
Marital status (reference = married)				
Single, divorced, and widowed	0.90 (0.82–0.99); 0.029	0.93 (0.85–1.02); 0.119	1.07 (0.98–1.17); 0.152	0.90 (0.82–0.99); 0.025
Employment (reference = employed)				
Unemployed	1.10 (0.99–1.22); 0.065	0.94 (0.85–1.04); 0.198	0.89 (0.81–0.99); 0.025	0.92 (0.83–1.02); 0.125
Profession (reference = other profession ^§^)				
Nutrition	0.94 (0.80–1.10); 0.431	1.00 (0.85–1.17); 0.983	0.94 (0.80–1.10); 0.431	1.25 (1.06–1.47); 0.008
Food	0.97 (0.86–1.11); 0.688	0.99 (0.87–1.12); 0.875	0.86 (0.76–0.98); 0.024	1.24 (1.08–1.42); 0.002
Agriculture	0.94 (0.76–1.15); 0.517	1.15 (0.95–1.41); 0.153	1.04 (0.85–1.27); 0.695	1.11 (0.90–1.37); 0.327
Sport	1.17 (0.92–1.47); 0.197	0.93 (0.74–1.16); 0.517	0.71 (0.57–0.89); 0.003	1.11 (0.88–1.41); 0.379
Psychology	1.06 (0.84–1.35); 0.628	1.13 (0.90–1.43); 0.288	0.89 (0.71–1.13); 0.337	0.94 (0.73–1.19); 0.589
Health	0.93 (0.83–1.04); 0.210	1.28 (1.15–1.43); <0.001	0.74 (0.67–0.83); <0.001	0.87 (0.77–0.98); 0.019
Following a healthy diet (reference = yes)				
Never or rarely follow	0.70 (0.62–0.79); <0.001	1.11 (0.99–1.24); 0.082	0.99 (0.89–1.12); 0.916	0.98 (0.87–1.11); 0.806
Body mass index (reference = obesity (BMI ^║^≥ 30.0)				
Underweight (BMI < 18.5)	0.77 (0.60–0.97); 0.030	1.59 (1.26–2.01); <0.001	2.19(1.73–2.77); <0.001	1.47 (1.15–1.87); 0.002
Normal weight (18.5 ≤ BMI ≤ 24.9)	1.34 (1.15–1.55); <0.001	1.30 (1.13–1.50); <0.001	1.55 (1.34–1.78); <0.001	1.24 (1.07–1.45); 0.005
Overweight (25.0 ≤ BMI ≤ 29.9)	1.26 (1.08–1.46); 0.003	1.17 (1.01–1.36); 0.034	1.13 (0.97–1.31); 0.105	0.94 (0.81–1.10); 0.457
Physical exercise (reference = weekly)				
Never exercise	0.64 (0.58–0.69); <0.001	1.30 (1.19–1.41); <0.001	0.80 (0.74–0.87); <0.001	0.88 (0.81–0.97); 0.007
Sitting in front of a television or computer (hours per day)	0.99 (0.98–1.01); 0.368	1.02 (1.00–1.03); 0.014	1.02 (1.00–1.03); 0.016	1.02 (1.00–1.03); 0.013
Motivation for health behavior (overall sum)	1.19 (1.18–1.21); <0.001	0.95 (0.94–0.96); <0.001	1.04 (0.86–1.26); 0.664	1.04 (1.04–1.05); <0.001
Emotional eating behavior (overall sum)	1.11 (1.10–1.12); <0.001	1.19 (1.19–1.20); <0.001	0.99 (0.82–1.21); 0.955	1.22 (1.21–1.22); <0.001

Note: * OR = adjusted odds ratio; ^†^ CI = (95% confidence interval; ^‡^
*p* = *p*-value; ^§^ Other profession = professions not related to the list; ^║^ BMI = body mass index.

## Data Availability

Not applicable.

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
