# Peer review of "Emotions and Food Consumption: Emotional Eating Behavior in a European Population"

_foods, 2023, doi:10.3390/foods12040872_

Round 1

Reviewer 1 Report

(check the wording of the title)

The authors conducted a cross-sectional study to assess associations between emotional state of eating, lifestyle, and emotional state in 9,052 people from 12 European countries. The authors used ordinal regression models to confirm associations between emotional state of eating and stress, depression, loneliness, boredom, and emotional comfort. The authors associated the emotional state of eating with an effort to improve physical and mental conditions such as weight control, staying awake and alert, and a general feeling of contentment. Comparisons were made between the 12 participating countries.

The study is appropriately and clearly designed, its scientific and technical quality are adequate, and it is well written and understandable. The following questions must be addressed:

1.- Introduction

Line 61-70. Please add a sentence that indicates a timeframe in which the events explained may take place (childhood, adolescence, old age, throughout life, etc.). This will help the reader better understand the scope of your study.

Line 71. Mention some examples of negative impacts on human health.

Could the authors add a sentence indicating why emotional eating overcomes the homeostatic control mechanisms that regulate hunger and satiety?

Line 100-101. Add salt, as well as sugar and fats.

Line 103. Delete the period before the reference [22].

2. Material and Methods

Check the numbering in this section.

By what criteria were the countries participating in the study selected? Was it simply for logistical reasons?

By what criteria were the participants in the different countries considered comparable? (For example, university population, company workers, etc.

Was the type of participating population standardized or was it a matter of random selection? This is not made clear in the manuscript. The average age of participants in Serbia is 23, for example, while in Hungary it is 43.

Please clarify how long it takes on average to fill out the questionnaire and where this activity was carried out. The instrument includes anthropometric data. The authors assume that the participants know their weight and height? Could you add it in the limitations section of the study?

Please state if any benefit was offered for participation in the study.

Do the authors consider the instrument used, validated in the Portuguese population, to be fully compatible with the population of the selected participating countries? Was any consistency test of the instrument performed for each of the participating countries?

Lines 164-165. Please expand on this explanation.

3. Results

Check the numbering of this section.

Review the end point of table 2.

Table 3 and 4. Change “yeas” to “years”

4. Discussion

Line 428. Add the word salt.

Line 436. To improve the scope of the discussion, it is suggested to comment on hyperpalatable foods with high energy density (excess sugar, fat and salt), which are generally ultra-processed, versus low-processed foods (fruits and vegetables). This should focus on consumer availability and health effects. This would considerably strengthen the nutrition and health aspects of the conclusions.

Author Response

Response to Reviewer 1 Comments

The authors conducted a cross-sectional study to assess associations between emotional state of eating, lifestyle, and emotional state in 9,052 people from 12 European countries. The authors used ordinal regression models to confirm associations between emotional state of eating and stress, depression, loneliness, boredom, and emotional comfort. The authors associated the emotional state of eating with an effort to improve physical and mental conditions such as weight control, staying awake and alert, and a general feeling of contentment. Comparisons were made between the 12 participating countries.

The study is appropriately and clearly designed, its scientific and technical quality are adequate, and it is well written and understandable. The following questions must be addressed:

Dear Reviewer, thank you very much for the opportunity to revise our manuscript. We appreciate your valuable comments for improvement our manuscript.

Thank you for your comment. We have responded to all of the points to the best of our abilities and we hope you will find them appropriate.

Point 1. 1.- Introduction

Line 61-70. Please add a sentence that indicates a timeframe in which the events explained may take place (childhood, adolescence, old age, throughout life, etc.). This will help the reader better understand the scope of your study.

Response 1. Thank you for your commnent. We added a few senteces: “These events may take place throughout life, but studies show that incidence of heightened emotionality increase in early adolescence, but in during all childhood with negative impact on eating pattern [5–7]. Although less than in younger people, emotional eating occurs in older people [5]. Women are more likely than men to develop eating disorders [8]. Thereby, excessive influence of social media may have strong influence in relationships between moods and eating and development of eating patterns [9].”

Point 2.

Line 71. Mention some examples of negative impacts on human health.

Response 2. Thank you for this reminder. We added a sentece: “An unbalanced diet may foster many chronic diseases such as metabolic syndrome, obesity, diabetes mellitus, hypercholesterolemia, high blood pressure, ischemic heart disease, stroke [11–15].”

Point 3.

Could the authors add a sentence indicating why emotional eating overcomes the homeostatic control mechanisms that regulate hunger and satiety?

Response 3. Thank you, we added the paragraph in introduction: “Also, emotional eating is the overlapping mechanism between reward circuitry, cognitive control, and emotions. Some of the pathophysiology mechanisms such as energy imbalance in hypothalamus and their relationship with ghrelin (“hunger hormone”) and leptin (“body's satiety signal”) have a significant impact on mood disorders [30]. Studies confirm that the consumption of highly palatable foods relates to pathways of the limbic system to mediate motivated behaviors, which explains why emotional eating overcomes the cognitive control mechanisms [31,32].

Point 4. Line 100-101. Add salt, as well as sugar and fats..

Response 4. This added.

Point 5.  Line 103. Delete the period before the reference [22].

Response 5. This corrected.

  1. Material and Methods

Point 6.  Check the numbering in this section.

Response 6. Thank you, this corrected.

Point 7.  By what criteria were the countries participating in the study selected? Was it simply for logistical reasons?

Response 7. In this article we were included only the data of the European Countries that effectively participated in the project EATMOT without dropping out. There was some condition to participate in the project, i.e., each country should collect a minimum number (500) of responses to the questionnaires. So, in total from 18 countries, 3 were out of Europa, and two countries failed to do this Cyprus and Macedonia didn't t deliver enough data to be included in the project/publications. So, all European Countries that actually collected data were included in this study.

Point 8. By what criteria were the participants in the different countries considered comparable? (For example, university population, company workers, etc.

Response 8. This study was intended to evaluate the behavior of the general population, not specific target groups. So the intention of the global project team and also passed on to each individual team in each of the courtiers was to collect responses from a minimum number of participants of about 500, and the only restriction was that they must be adults. All other characteristics would be random, since we used a convenience sample. The project did not provide enough funds or even enough time to undertake the research on a stratified sample, with equivalent number of participants for each of the categories of the sociodemographic variables in each of the countries. And that was not the first option at all also due to logistic constrains in each of the countries. The project was approved under the conditions of having a convenience sample, which is even often recommended for studies that aim to mimic the behavior of the general consumer. In consumer studies, unless we are dealing with products specifically destined to special groups like sports practitioners or elderly, for example, in every other situation, the use of convenience samples is adapted to the reality of buying. If one goes to the entrance of two different supermarkets on any day, and enquires the type of clients, it will not be obtained a similar sample in terms of age, education level, sex, etc… and that is the reason why authors consider that the use of convenience samples, although having some limitations, is reasonable for this type of study, and allows valuable conclusions that are valid for the general population. A quick search on the ScienceDirect website under the keywords “convenience sample” today resulted in 321,887 (almost four hundred thousand publications), many of them describing studies undertaken on convenience samples, precisely for their availability, easiness of recruitment, and relatively good adaptation to the reality, that tends to be random by nature.

Point 9. Was the type of participating population standardized or was it a matter of random selection? This is not made clear in the manuscript. The average age of participants in Serbia is 23, for example, while in Hungary it is 43.

Response 9. The sample was a convenience sample, for operational reasons. So population was randomly selected but all included participants had to be over 18 years old in each country.

Point 10. Please clarify how long it takes on average to fill out the questionnaire and where this activity was carried out.

Response 10. Thank you for this reminder. We added sentences in paragraph 2.2: “Filling out the questionnaire took an average of 15 minutes. Both the location and the time for filling were up to the participants. The researchers received the completed questionnaires by courier or in person.”

Point 11. The instrument includes anthropometric data. The authors assume that the participants know their weight and height? Could you add it in the limitations section of the study?

Response 11. Thank you for your comment. We added sentence inn limitation: “Third, used the self-assessment method to assess weight and height, which may affected the objectivity of data.“

Point 12.  Please state if any benefit was offered for participation in the study.

Response 12. We added the sentence:The study was voluntary, and was not any compensation for research participants.“ in paragraph 2.1.

Point 13. Do the authors consider the instrument used, validated in the Portuguese population, to be fully compatible with the population of the selected participating countries? Was any consistency test of the instrument performed for each of the participating countries?

Response 13. Than you for this comment. We describer already and now added the sentenceFor each of the participating countries, were conducted a small pilot test on ten respondents on different age groups before administering the questionnaire to ensure that it was understandable and would yield results that fell within the expected range.”

Point 14.  Lines 164-165. Please expand on this explanation

Response 14. Thank you for this comment. We deleted this sentence, and added: “The validation of questionnaire was described in our previous papers [38,39]. The validated instrument guarantees confidence and confirm that information obtained through this instrument are compatible with the population of the selected participating countries [39].”

  1. Results

Point 15.  Check the numbering of this section.

Response 15. Thank you, this corrected.

Point 16. Review the end point of table 2.

Response 16. This corrected.

Point 17. Table 3 and 4. Change “yeas” to “years”

Response 17. This corrected.

  1. Discussion

Point 18.  Line 428. Add the word salt.

Response 18.  This added.

Point 19.  Line 436. To improve the scope of the discussion, it is suggested to comment on hyperpalatable foods with high energy density (excess sugar, fat and salt), which are generally ultra-processed, versus low-processed foods (fruits and vegetables). This should focus on consumer availability and health effects. This would considerably strengthen the nutrition and health aspects of the conclusions.

Response 19. Thank you for your comment. We added the paragraph:Awareness of the significance of dietary fiber foods within dietary patterns provides consumers to choose healthy in lieu of unhealthy food [49–51]. Hyperpalatable foods, which are typically ultra-processed and abundant with high energy density, excess sugar, fat, and salt, as well as meat and animal products, need to be substituted with fruits and vegetables, like legumes, in order to prevent detrimental impacts on health [51–53]. This food is nutritious, high-protein, and nutrient-dense foods that can avoid a number of chronic diseases such as heart disease, stroke, diabetes, bowel cancer, obesity [53]. Despite that, stress may contribute to the listlessness, social isolation, fatigue and less physical activity. In addition to downregulating the cognitive control centers of the brain, stress activates "reward pathways", and increases cravings for palatable foods. This com-bination of neural adaptations frequently leads to an increase in the consumption of palatable foods, such as comfort foods high in fat and/or sugar [54].”

Reviewer 2 Report

Thank you very much for the opportunity to read the text.
I would like to draw the authors' attention to some moments that should be improved.
For example, in the Abstract, the aim of the study should be stated at the beginning, the most important results should be given at the end (they should be listed, not statistical values - if the abstract interests the reader, he will reach for the whole article).
From the Introduction, too, issues relating to future research should be moved to the appropriate subsection at the end.

In the methodology section, the formulae used should be stated. And why the authors chose to use regressions.

The section numbering is broken. The Results section is numbered 1. It should be numbered 3.

On what principle was the selection of countries for the study. What was the key to the selection.

The results section is an accumulation of data as in the text as in the tables. Interpretations of what came out in the study should be given. Such a presentation without one's own analysis and thoughts discourages one from reading such a work.

The LIMITATIONS subsection is missing from the final section.

The discussion could still refer to studies of healthy eating habits by authors such as:
Fedeli, R.; Reyneke, G.; Śmiglak-Krajewska, M.; Pucci, T.; Carfora, V.; Celetti, S.
The indicated authors have recently published very interesting works on consumer behaviour and healthy habits, e.g. replacing unhealthy foods with healthy ones.

It would be useful to contrast your results with their summaries of own research.

The article uses 50% of the literature that is older than 5 years.

Author Response

Response to Reviewer 2 Comments

Thank you very much for the opportunity to read the text.

I would like to draw the authors' attention to some moments that should be improved.

Dear Reviewer, thank you very much for the opportunity to revise our manuscript. We appreciate your valuable comments for improvement our manuscript. Thank you for your comment. We have responded to all of the points to the best of our abilities and we hope you will find them appropriate.

Point 1. For example, in the Abstract, the aim of the study should be stated at the beginning, the most important results should be given at the end (they should be listed, not statistical values - if the abstract interests the reader, he will reach for the whole article).

Response 1. Thank you for you comment. We deleted first two sentences, deleted some of statistical data, and added main important results at the end of abstract: “The emotional state of eating was associated with an effort to improve physical and mental conditions such as weight control, stress relief, depression, staying awake and alert, and a general feeling of contentment and consolation”

Point 2.  From the Introduction, too, issues relating to future research should be moved to the appropriate subsection at the end.

Response 2. We replaced this sentence.

Point 3. In the methodology section, the formulae used should be stated. And why the authors chose to use regressions.

Response 3. Thank you for your comment. We already describe the formulas of BMI in methods “Participants' BMI was calculated using the formula: BMI=body weight (kg)/(body height (m2) (BMI= kg/m2)) from self-assessed anthropometric data (weight in kg, height in m).”, an explain why we used regression “We created multivariate regression models to assess the association between the emotional status and reason for food consumption, motivation for healthy nutrition behavior, and predictors

Point 4.  The section numbering is broken. The Results section is numbered 1. It should be numbered 3.

Response 4. This corrected.

Point 5. On what principle was the selection of countries for the study. What was the key to the selection?

Response 5. In this article we were included only the data of the European Countries that effectively participated in the project EATMOT without dropping out. There was some condition to participate in the project, i.e., each country should collect a minimum number (500) of responses to the questionnaires. So, in total from 18 countries, 3 were out of Europa, and two countries failed to do this Cyprus and Macedonia didn't t deliver enough data to be included in the project/publications. So, all European Countries that actually collected data were included in this study.

Point 6. The results section is an accumulation of data as in the text as in the tables. Interpretations of what came out in the study should be given. Such a presentation without one's own analysis and thoughts discourages one from reading such a work.

Response 6. Thank you for this comment. We shortened the text in this section. We attempted to describe only the information that was significant and relevant to our goal. We tried to describe the association of very complex variables. We are aware of the complex text, but our display of results here is very elaborate in order to better explain our unique situation. We understand your point of view, but the journal requirements are to have a section just for the results separated from the discussion. In this way in the results section it is not intended to make any discussion of the results at all, or even use analysis of the results or justifications or comparisons with other studies published in the scientific literature, because the discussion is in the following section named exactly that. It is debatable but the authors did follow in fact the structure of the template for this journal. For that reason, we shortened the text in the results part, only to evidence the results, not discussing them. Furthermore, we tried to describe mostly and give emphasis to data which was significant and relevant for our particular aim. We tried to describe associations of very complex variables. We are aware of some possible complexity of the text, and even some statistical treatment undertaken, but they are relevant for the purpose of the research and the discussion part focused on the general analysis of the results, form a macro-perspective, i.e., focusing on the whole purpose of this research focusing specifically on citizens from European countries.

Point 7.  The LIMITATIONS subsection is missing from the final section

Response 7. We added this.

Point 8. The discussion could still refer to studies of healthy eating habits by authors such as: Fedeli, R.; Reyneke, G.; Śmiglak-Krajewska, M.; Pucci, T.; Carfora, V.; Celetti, S. The indicated authors have recently published very interesting works on consumer behaviour and healthy habits, e.g. replacing unhealthy foods with healthy ones. It would be useful to contrast your results with their summaries of own research.

Response 8. Thank you for you improve of discussion. We cited these authors in discussion.

Point 9. The article uses 50% of the literature that is older than 5 years.

Response 9. We corrected this. Thank you for your reminder.

Round 2

Reviewer 2 Report

Thank you very much for the opportunity to read the article with the amendments made by the authors.
The authors have given maximum consideration to all the suggested changes.
I do not have any more suggestions to correct.
I wish you the best of luck in your future research.

Author Response

Thank you very much for the opportunity to revise and additionally improve our manuscript. We appreciate your valuable comments for improvement our manuscript. 
Thank you for all your comments.  Whole article was edited by editing services listed at https://www.mdpi.com/authors/english.